# Prader–Willi Syndrome with Angelman Syndrome in the Offspring

**DOI:** 10.3390/medicina57050460

**Published:** 2021-05-08

**Authors:** Donatella Greco, Luigi Vetri, Letizia Ragusa, Mirella Vinci, Angelo Gloria, Paola Occhipinti, Angela Antonia Costanzo, Giuseppe Quatrosi, Michele Roccella, Serafino Buono, Corrado Romano

**Affiliations:** 1Oasi Research Institute-IRCCS, 94018 Troina, Italy; dgreco@oasi.en.it (D.G.); lragusa@oasi.en.it (L.R.); mvinci@oasi.en.it (M.V.); agloria@oasi.en.it (A.G.); pocchipinti@oasi.en.it (P.O.); acostanzo@oasi.en.it (A.A.C.); fbuono@oasi.en.it (S.B.); cromano@oasi.en.it (C.R.); 2Department of Sciences for Health Promotion and Mother and Child Care “G. D’Alessandro”, University of Palermo, 90128 Palermo, Italy; giuseppe.quatrosi01@community.unipa.it; 3Department of Psychology, Educational Science and Human Movement, University of Palermo, 90128 Palermo, Italy; michele.roccella@unipa.it

**Keywords:** Angelman syndrome, fertility, offspring, Prader–Willi syndrome

## Abstract

We report the second case, to the best of our knowledge, of a mother with Prader–Willi syndrome (PWS) who gave birth to a daughter with Angelman syndrome (AS). The menarche occurred when she was 16, and the following menstrual cycles were irregular, but she never took sexual hormone replacement therapy. At the age of 26, our patient with PWS became pregnant. The diagnosis was confirmed by molecular genetic testing that revealed a ~5.7 Mb deletion in the 15q11.1–15q13 region on the paternal allele in the mother with PWS and the maternal one in the daughter with AS, respectively. Both the mother with PWS and the daughter with AS showed peculiar clinical and genetic features of the two syndromes. Our case report reaffirms the possible fertility in PWS; therefore, it is very important to develop appropriate socio-sexual education programs and fertility assessments in order to guarantee the expression of a healthy sexuality.

## 1. Introduction

Chromosome 15q11–q13 region harbors several genes regulated by genomic imprinting; therefore, they are functionally haploid. Prader–Willi syndrome (PWS) and Angelman syndrome (AS) are rare genetic neurodevelopmental conditions provoked by lack of expression of imprinted genes in the 15q11–q13 critical region.

It has become clear that loss of function of the UBE3A gene, expressed only on the maternal chromosome, causes AS. Several genetic mechanisms may cause a functional impairment of maternal inherited UBE3A gene: a mutation on the maternal allele (5–10% of cases), a deletion of the maternally inherited chromosomal critical region (about 75% of cases), chromosome 15 paternal uniparental disomy (1–2% of cases), or an imprinting center defect (1–3% of cases) [1].

In PWS the lack of expression of the paternally derived chromosome, 15q11–q13, can occur in one of three ways: paternal 15q11–q13 deletion (65–75% of cases), maternal uniparental disomy 15 (20–30% of cases), and imprinting center defect (1–3%) [2].

Deficiency of the paternally expressed SNORD116 is sufficient to cause the most salient features of PWS, emerging, therefore, as a possible candidate gene in PWS. However, we cannot exclude that other paternally expressed genes in the 15q11–q13 critical region contribute to the full phenotype.

PWS clinical features include earlier hypotonia and feeding difficulties, and later on they include developmental delay, hyperphagia with subsequent obesity, distinctive facial features (narrow forehead, almond-shaped eyes, thin upper lip, downturned mouth), very small hands and feet, short stature, behavioral and psychiatric disturbances, epilepsy, and hypogonadotropic hypogonadism with delayed or incomplete pubertal development, determining a frequent infertility. AS clinical features include severe intellectual disability, microcephaly and distinctive facial features, speech impairment or absence of language, epilepsy, characteristic electroencephalographic abnormalities, abnormal sleep patterns, hyperactivity, unprovoked episodes of laughter and smiling, jerky movements, and tremors and ataxia [3,4,5,6,7].

Here we report on a woman with PWS giving birth to a daughter with AS.

## 2. Case Report

### 2.1. Materials and Methods

The molecular diagnosis of PWS was achieved through methylation test of SNRPN gene (through chemical DNA modification followed by methylation specific PCR–MSP), documenting the single maternal methylated allele, compatible with PWS. The diagnosis was confirmed by multiplex ligation-dependent probe amplification (MLPA) analysis (by SALSA MLPA Kit ME028B1 Prader Willi/Angelman) that allowed for the contemporary possibility of assessing the methylation status and the presence of alterations in the number of copies in the PWS/AS critical region. The MLPA showed the absence of paternal alleles including MKRN3, MAGEL2, NDN, SNRPN, UBE3A, ATP10A, GABRB3 genes, and HBII-85 snoRNA cluster in the 15q11.1–15q13 chromosomal region. The MLPA analysis was compatible with PWS diagnosis due to ~5.7 Mb paternal deletion in the 15q11.1–15q13 region (Figure 1).

The AS was diagnosed through the methylation test, that showed the single paternal methylated allele compatible with AS. Moreover, the diagnosis was confirmed through the molecular analysis of MLPA, showing the absence of maternal alleles including MKRN3, MAGEL2, NDN, SNRPN, UBE3A, ATP10A, GABRB3 genes, and HBII-85 snoRNA cluster in the 15q11.1–15q13 chromosomal region. The MLPA analysis was compatible with AS diagnosis due to ~5.7 Mb maternal deletion in the 15q11.1–15q13 region (Figure 1).

This study was approved by the local ethics committee “Comitato Etico IRCCS Sicilia-Oasi Maria SS.” on 16 June 2020, approval code: 2020/06/16/CE-IRCCS-OASI/34.

### 2.2. The Mother with Prader–Willi Syndrome

The diagnosis of PWS was confirmed at the age of 8 by the molecular genetics test underlying a 5.7 Mb deletion of paternal origin in the PWS/AS critical region. Moreover, the clinical diagnostic criteria according to Holm1 were fully satisfied. The family history was negative either for PWS or AS or other genetic, neurological, and psychiatric conditions.

The patient presented obesity with BMI = 39.4 kg/m^2^ and type 2 diabetes mellitus. The girl developed a typical PWS behavioral phenotype, characterized by obsessive-compulsive behaviors, rigid thinking, suspiciousness, low frustration tolerance, emotional vulnerability, and self-destructive behaviors, like skin picking. Therefore, she is treated with Risperidone and Oxcarbazepine. In the cognitive domain we documented a mild intellectual disability with an uneven profile of the WAIS-IV scores (normal verbal subtests, deficits in performance subtests). The patient was affected by a deficit in the growth hormone (GH), hypothyroidism, and type 2 diabetes mellitus. Therefore, the patient was receiving treatment with L-thyroxine, insulin therapy (Humalog 4U; Tresiba 4U), and GH replacement therapy from 6 to 18 years old.

The menarche occurred when she was 16, with irregular menstrual cycles following. She never took sexual hormone replacement therapy. The gynecological ultrasound showed an anteverted uterus of regular size and eco structure, and normal bilateral ovaries. At the age of 26, following a physical abdominal examination, a pelvic ultrasound and a pregnancy test were performed. Gestation, through fetal ultrasonography, was estimated to be at 28 weeks. Her partner had no neurological, psychiatric, or genetic disorders. The pregnant woman and her parents were informed about a 50% chance that the fetus was affected by AS.

Since in PWS there are a reduction of pain threshold and behavioral problems, and assuming the risk of lack of cooperation during labor, the caesarean section at 38 weeks of gestation was agreed upon and planned.

The mother was not able to take care of her daughter, she did not breastfeed her and, although they lived together, the baby was always left in the care of her maternal grandparents.

### 2.3. The Daughter with AS

The daughter with AS was born after a 36-week pregnancy from an emergency caesarean section performed for premature labor. At birth, considering the high risk of offspring with AS from mother with PWS, and considering the axial hypotonia, we proceeded to conduct molecular genetics tests that confirmed the diagnosis of AS.

Weight at birth was 2495 g, length 47 cm, head circumference 32 cm, APGAR: 5′:6; 10′:8. At birth she presented with respiratory distress and hypoglycemia. The cardiac echo at birth showed patent foramen ovale with mild left to right shunt and patent ductus arteriosus with continuous left to right shunt and medium/high-speed flow patterns. The brain scan showed subependymal cysts to the caudothalamic grooves of both lateral ventricles, and mild periventricular hyper echogenicity.

At three months the follow-up showed that the head circumference was 37 cm (2nd percentile), head control was almost reached, social smile was present, and generalized hypotonia and rightward convex nasal septal deviation were also detected. At 10 months the child seemed to have discrete relational and environmental skills. The main clinical features at this age were global developmental delay, good trunk control—she sat without any support, but she was not able to crawl or stand with support—absence of speech, babbling or spontaneous use of gestures, microcephaly and brachycephaly, protruding tongue, generalized muscular hypotonia and hyperreflexia, and gastroesophageal reflux symptoms. The hematochemical values of routine blood exams were normal, as well as the thyroid hormone levels.

An electroencephalogram (EEG) performed at 11 months showed widespread polymorphic theta-delta activity, predominantly on the posterior regions of the two hemispheres.

## 3. Discussion

Infertility is a constant feature of both male and female individuals affected by PWS, because sexual maturity does not fully develop in PWS. In people with PWS puberty is frequently delayed and/or incomplete. However, they have a premature pubarche characterized by earlier appearance of pubic hair. In girls there is mostly some breast development and in boys a degree of penile growth. Cryptorchidism, small testes, scrotal hypoplasia, and a micro-penis, are often present in boys. Hypoplasia of the clitoris, primary or secondary amenorrhea, and late menarche (frequently after their 20s) are often present in girls. Menstruation, when present, is often irregular and feeble, but some girls can have regular menstrual cycles [8,9].

Congenital hypogonadism is responsible of the complete or partial pubertal failure and seems to have a mixed etiology in males. Central forms with relatively low LH levels, peripheral forms with low inhibin B and relatively high FSH levels, and combined forms have been reported [10,11].

There is literature evidence that primordial follicle pool and small antral follicles are conserved in girls with PWSm and they do not show a classical hypogonadotropic hypogonadism. However, the progression of pubertal development is impaired, probably because of low levels of LH, while anti-Mullerian hormone and FSH are normal and inhibin B levels are in the low-to-normal range. The timing and the peak of LH play a key role in ovulation, and thus, fertility. In case of no LH dysregulation, normal inhibin B levels and regular menses, contraception should be considered in women with PWS [12]. Secondary sexual female characters have a variable development, and some studies show complete Tanner V breast development [13].

Since these impairments are not absolute, ovulation and conception cannot be completely excluded in girls with PWS. The complex pattern analyzed in male adolescents, the combination of a primary gonadal defect and hypogonadotropic hypogonadism, seems to be similar to gonadal dysfunction in PWS females [14,15].

Nevertheless, with sex hormone treatment, both girls and boys will develop some physical maturity. The degree of hypogonadism varies in severity, and because of it, pubertal development is absent in some and delayed or arrested in others [2,16]. However, there is no consensus on the more appropriate regimen of sex hormone treatment in PWS. The hormone treatment is mainly influenced by the age at diagnosis and by local practices. The main protocols included a therapeutic hCG treatment and testosterone replacement improving cryptorchidism, and virilization, leading to an improvement in the quality of life in males with hypogonadism. The hormonal replacement therapy in females with PWS often consists of oral estrogen alone or in combination with progestin, which are usually well tolerated [17,18,19].

In a recent review, Noordam et al. clearly affirmed that the evidence on hypogonadism in PWS and its treatment is limited, and it is often based on expert opinion. They suggest starting sex hormones substitution at the same age and using the same dosage regimen used for normal hypogonadal children and adolescents. However, much evidence is needed, in particular, to clarify the effects of the long-term treatment on muscle mass and peak bone mass in order to extend the supplementation of sex steroids in PWS [20].

Understanding the specific genetic causes of PWS is fundamental for proper genetic counseling because the risk of recurrence depends on the causal mechanism. The three main mechanisms leading to PWS are paternal microdeletion, maternal uniparental disomy, and imprinting defects; occasionally deletion results from chromosomal translocation. Due to a greater risk of recurrence, fathers of children with microdeletion should have the opportunity to undergo the FISH analysis or MLPA of the 15q11.2–q13 region. In the case of maternal uniparental disomy, the analysis of microsatellite markers in the critical region 15q11.2–q12 for the mother, father, and proband allows for the determination of the inheritance of alleles as biparental or uniparental, detecting both uniparental heterodisomy and uniparental isodisomy. If the analysis of microsatellite markers is normal, a Robertsonian translocation in the father should be excluded. Most imprinting defects are caused by an epigenetic mutation with a recurrence risk of less than 1%, in rare cases there may be a microdeletion of the imprinting center, which can be familiar with a recurrence of 50% [21].

The genetic type of PWS influences the chances of a mother with PWS having a healthy baby. A woman with maternal disomy is likely to have a healthy baby. A woman with a deletion has a 50% chance of having a healthy baby and a 50% one of having a baby with AS. A man with maternal disomy might be the father of a healthy child. A man with a PWS deletion has a 50% chance of fathering a baby with PWS and has a 50% chance of having a healthy baby.

There are no known cases in literature, to date, of men with PWS fathering a child, while five females diagnosed with PWS were reported to have delivered (Table 1). Three cases were reported before that molecular PWS diagnosis was fully developed. In the first two cases, the diagnosis was exclusively clinical without confirmation by cytogenetic or molecular genetic technique. One of these mothers gave birth to four children, and among them, one had the clinical suspicion of PWS; the other mother had a single pregnancy mothering a healthy baby [22].

In another report a woman with a cytogenetically identified deletion of region 15q11–q13 gave birth to two children: a healthy baby and one inheriting the deletion and therefore having a possible AS diagnosis [23,24]. However, the deletion present in this family lacked a molecular confirmation and, on this basis, the PWS diagnosis was questioned [25].

In a Swedish case report, a woman with molecularly confirmed PWS due to a uniparental disomy (UPD) was reported to have given birth to an unaffected girl. This woman received the hormone treatment with medroxyprogesterone and took selective serotonin reuptake inhibitors (SSRIs). The authors suggested that these treatments might have stimulated the gonadotropin release and facilitated ovulation and pregnancy [26].

In the more recent reported case by Schulze et al., a 32-year-old woman with PWS gave birth to a girl with AS. The mother has a paternally inherited deletion of the 15q11–q13 chromosome that provoked PWS, while her daughter inherited her mother’s mutation; therefore, she is affected by AS. The deletion in both mother and daughter was detected by karyotype and FISH and was confirmed with PCR-based microsatellite analysis. Unlike the previous case, no hormone replacement therapy, contraception, or any other treatment was received by the mother [27]. The phenotype of the girl with AS reported by Schulze et al. was deeply described from early childhood to adolescence by J. R. Ostergaard. The girl showed a typical AS phenotype during early childhood characterized by jerky movements, poor sleep, typical electroencephalography pattern, epilepsy, severe developmental disability, outbursts of laughter, and characteristic dysmorphic facial features such as a round face, protruding tongue, pointed cheeks, wide mouth, and mandibular prognathism. Interestingly, dysmorphic traits disappeared when she grew older and typical behavioral disturbances of PWS increased, such as social withdrawal in unfamiliar surroundings, insistence on rigid routines, hyperphagia, and a body fat distribution similar to that reported in patients with PWS [28].

## 4. Conclusions

People with genetic disorders have a greater risk of experiencing sexual development disorders and fertility issues. In PWS sexuality and fertility are inevitably conditioned by biological deficits such as hypogonadism that determines a reduced production of sexual hormones and pubertal alterations. These factors reduce fertility and lead people with PWS to feel and look different from their peers [29,30]. Nevertheless, it is important the psychological aspect of the PWS often characterized by pathological features such as obsessive–compulsive behavior, mood disorders, and impulsivity that can lead to inappropriate social and sexual behaviors as a result of those problems, as well as to social skill deficits [31].

It is therefore very important, considering the possible fertility in PWS reaffirmed by our case report, to develop appropriate socio-sexual education programs aimed to guide people with PWS and their caregivers to a higher awareness of affective and sexual practices characterized by a better sense of self, both mentally and bodily, and by the perception of being responsible for one’s feelings, desires, and actions, as well as an evaluation of fertility condition to assess possible contraception methods.

Preparatory courses for families and for people suffering from the syndrome should be organized with personnel specialized in sexology, but who are well aware of the problems related both to intellectual disability and to the characteristics of the behavioral phenotype of Prader–Willi syndrome.

This is necessary to guarantee a meaningful realization of human rights, in people with PWS or other genetic syndromes, that cannot be separated by the expression of a healthy sexuality [32].

## Figures and Tables

**Figure 1 medicina-57-00460-f001:**
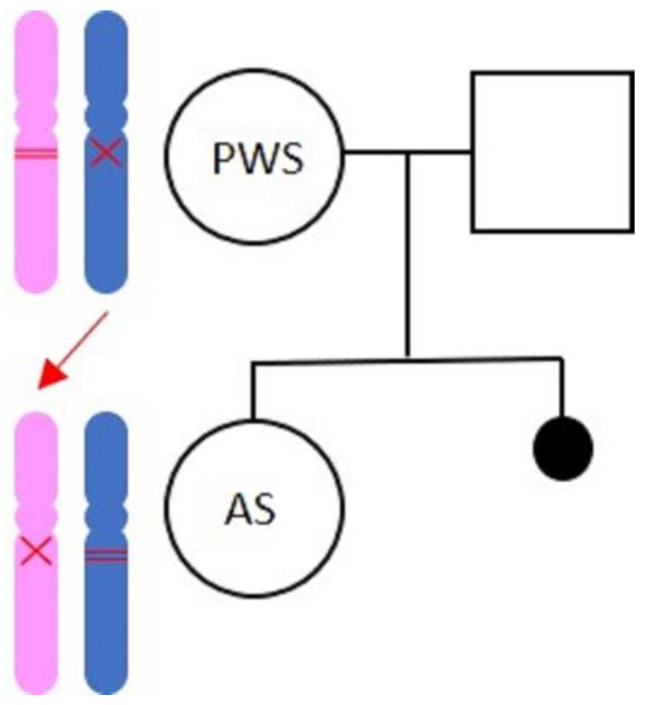
Genetic family tree. In the mother, the paternal allele got deleted and the maternal allele is methylated (condition compatible with PWS). The daughter inherits the deleted maternal allele, and the paternal allele is methylated (condition compatible with AS).

**Table 1 medicina-57-00460-t001:** Mothers with Prader–Willi syndrome.

Case	Diagnosis	Deliveries	Age at Delivery	Healthy Children	Affected Children	Reference
1	Clinical	4	29 y	1	1: clinical PWS 2: congenital heart defects	Laxova et al., 1973 [22]
2	Clinical	1	28 y	1		Laxova et al., 1973 [22]
3	Cytogenetical (deletion)	2	Unknown	1	1: precocious puberty	Hockey et al., 1986 [23] Hockey et al., 1987 [24]
4	Molecular (UDP)	1	33 y	1		Åkefeldt et al., 1999 [26]
5	Molecular (deletion)	1	32 y		1: Angelman syndrome	Schulze et al., 2001 [27]

## Data Availability

Data available under request to the corresponding author of the study.

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
