# Peer review of "Prader–Willi Syndrome with Angelman Syndrome in the Offspring"

_medicina, 2021, doi:10.3390/medicina57050460_

Round 1

Reviewer 1 Report

This is a case report on a woman with PWS mothering a child. The authors emphasize the possibility of fertility in PWS and describe genetic and pathophysiological backgrounds.

The Case report is interesting but needs some further rewriting and attention.

Specific comments on content:

Could you include a comment on the family history. More cases of PWS/AS in the family ? Add a comment on proper genetic counseling of parents of a child with PWS.

I do miss a recommendation to the reader how to judge fertility. Include the use of measurements of Inhibin-B.

The authors use the term socio-sexual education. What do they mean by this and who should be the target for this education ?

Text:

2.1 how was the situation of the child ? was she living with her parents or in a residence ? Role of caretakers ?

sentence 49: "The patient presents with severe obesity ...." : not clear what is meant.

Line 55 "GH" please write in full

Line 57: Insulin dosages are very very low. Are these correct ?

2.2 on the daughter: description of neonatal course should be shortened. Does not contribute to the case.

3. Discussion

The part on (in)fertility/hypogonadism and puberty is not very well written. Please rewrite.

Line 120-122: not clear and not correct. The children either have Precocious puberty or premature pubarche. The last without penile enlargement. Line 140-143 is on the same topic. Please connect these parts and rewrite.

Include comment on measurement of Inhibin-B to get an impression of fertility.

I suggest to include comment on instruction/ education of the care takers on sexuality.

Author Response

Dear Reviewers,

I would like to thank you for your valued comments and suggestions to the article. As you requested, we made all the necessary changes in our manuscript to address your concerns and we detailed below how the points raised have been accommodated. The main changes are written in red in the text of the manuscript. From the changes made in the revised manuscript and responses provided below, I hope you are convinced that we have adequately addressed the reviewer’s concerns and made the paper better. If there are any further questions, please feel free to let me know.

This is a case report on a woman with PWS mothering a child. The authors emphasize the possibility of fertility in PWS and describe genetic and pathophysiological backgrounds.The Case report is interesting but needs some further rewriting and attention.

Specific comments on content:

Could you include a comment on the family history. More cases of PWS/AS in the family?

Thanks for the suggestion. The family history was negative for PWS or AS or other genetic, neurological and psychiatric conditions. We have added this clarification in the text (lines 83-84)

Add a comment on proper genetic counseling of parents of a child with PWS.

Thanks for the suggestion. We added some information about proper genetic counseling for parents with a child with PWS (please see lines 181.193).

I do miss a recommendation to the reader how to judge fertility. Include the use of measurements of Inhibin-B.

Thanks for the suggestion. We provided to the reader more details about how to judge fertitily, including the use of inhibin-B (please see lines 149-155)

The authors use the term socio-sexual education. What do they mean by this and who should be the target for this education?

Thanks for the suggestion. We better explained what we mean by socio-sexual education and its target (please see lines 247-256)

Text:

2.1 how was the situation of the child? was she living with her parents or in a residence? Role of caretakers ?

Thanks for the suggestion. The mother was not able to take care of her daughter, she did not breastfeed her and although they have been living together, the baby has always been left in the care of her maternal grandparents (please see lines 106-108).

sentence 49: "The patient presents with severe obesity ...." : not clear what is meant.

The patient has a BMI of 39.4 kg/m2 near class III obesity. We deleted the adjective severe.

Line 55 "GH" please write in full

Thanks for the suggestion. We corrected it.

Line 57: Insulin dosages are very very low. Are these correct?  

Yes, they are. They are enough for her glucose homeostasis.

2.2 on the daughter: description of neonatal course should be shortened. Does not contribute to the case.

Thanks for the suggestion. We summarized it.

  1. Discussion

The part on (in)fertility/hypogonadism and puberty is not very well written. Please rewrite.

Thanks for the suggestion. We rewrote this part (please see lines 136-144)

Line 120-122: not clear and not correct. The children either have Precocious puberty or premature pubarche. The last without penile enlargement. Line 140-143 is on the same topic. Please connect these parts and rewrite.

Thanks for the suggestion. We made the concept clearer (please see lines 136-144)

Include comment on measurement of Inhibin-B to get an impression of fertility.

Thanks for the suggestion. We added a comment in lines 151-155

I suggest to include comment on instruction/ education of the care takers on sexuality.

Thanks for the suggestion. We added in the conclusion paragraph comment on instruction/ education of the caretakers on sexuality (lines 247-256).

Reviewer 2 Report

In this case report dr. Greco et al. describe the genetics of a woman with PWS, who became pregnant. A girl with genetically confirmed AS was born by emergency caesarean section, and the girl’s development during 11 months after delivery is reported.

This case report adds to the limited knowledge of fertility in PWS and is of great interest and importance for clinicians taking care of these patients. However, more clinical information would be of great value and would increase the novelty of the case report.

Other comments:

  1. No information of the father is mentioned. If available, it would be of great to include or just mention that no information about him was available.
  2. In line 55, it says the mother had a GH deficiency. Was it ever replaced?
  3. The pregnancy was noticed in gestational week 28, line 63. Was the L-thyroxine dose increased? Were the thyroid hormones low and in that case would that have affected the girl?
  4. Was the metabolic control good during pregnancy?
  5. Was the mother breast feeding the baby?
  6. Was the mother able to take care of the baby herself and to bond with her?
  7. Line 148: Some suggestions how to treat hypogonadism in PWS can be found in a recent review by Noordam C et al. (2021) and could be mentioned/referred to.
  8. There is a follow-up case report of the Danish girl with AS (Ostergaard J 2015) which could be included in the discussion section.

Author Response

Dear Reviewers,

I would like to thank you for your valued comments and suggestions to the article. As you requested, we made all the necessary changes in our manuscript to address your concerns and we detailed below how the points raised have been accommodated. The main changes are written in red in the text of the manuscript. From the changes made in the revised manuscript and responses provided below, I hope you are convinced that we have adequately addressed the reviewer’s concerns and made the paper better. If there are any further questions, please feel free to let me know.

Reviewer 2

In this case report dr. Greco et al. describe the genetics of a woman with PWS, who became pregnant. A girl with genetically confirmed AS was born by emergency caesarean section, and the girl’s development during 11 months after delivery is reported.

This case report adds to the limited knowledge of fertility in PWS and is of great interest and importance for clinicians taking care of these patients. However, more clinical information would be of great value and would increase the novelty of the case report.

Other comments:

  1. No information of the father is mentioned. If available, it would be of great to include or just mention that no information about him was available.

Thanks for the suggestion. Her partner has no neurological, psychiatric or genetic disorders. No other details have been provided by the family.

2. In line 55, it says the mother had a GH deficiency. Was it ever replaced?

Yes, she received GH replacement therapy from 6 to 18 years old. We specified it in the text (line 94)

3. The pregnancy was noticed in gestational week 28, line 63. Was the L-thyroxine dose increased? Were the thyroid hormones low and in that case would that have affected the girl?

Thyroidal hormone levels were normal during pregnancy.

4. Was the metabolic control good during pregnancy?

Yes, the metabolic control and in particular the glucose metabolism were good during pregnancy.

5. Was the mother breast feeding the baby?

No, the mother did not breastfeed the baby. We specified it in the text (please see lines 106-108)

6. Was the mother able to take care of the baby herself and to bond with her?

No, but she lives with the baby, we added more details about it, please see lines 106-108

7. Line 148: Some suggestions how to treat hypogonadism in PWS can be found in a recent review by Noordam C et al. (2021) and could be mentioned/referred to.

Thanks for the suggestion. We cited the article and added more information about how to treat hypogonadism in PWS (please see lines 175-180)

8. There is a follow-up case report of the Danish girl with AS (Ostergaard J 2015) which could be included in the discussion section.

Many thanks for the suggestion, we reported the findings of follow-up case by Ostergaard J 2015 in the discussion. Please see lines 224-233.

Reviewer 3 Report

Manuscript concerned rare genetic syndromes and extremely rare possibility of giving birth by persons with Prader-Willi syndrome (PWS). This is interesting and good presented case report, which focused our attention on new aspect of care and sexual education patients with this syndrome. But I think, a few points should be more developed or added.

  1. I think that genetics of PWS, and problem of maternal imprinting should be more explain in the introduction.

PWS is human genetic imprinting disorder that can be caused by paternal deletion, maternal uniparental disomy or imprinting centre defect, not only this what was described.

  1. The way, how Angelman Syndrome (AS) develops should be showier expound.
  2. There are too small about characteristic features of AS children.
  3. Minor problems Line 31-32- I suggest replace the sentence from: “It has become clear that loss of function of the UGE3A gene, expressed on the maternal chromoseome, only causes AS” to “…………, expressed only on the maternal chromosome, only causes AS. What definitely change the meaning.
  4. I do not know, if it is necessary to write about PWS-like syndrome.
  5. Lines 102-103- this sentence should be changed- trunk control is not the cause of GERD, what is suggested by this sentence.
  6. Despite quite long follow up of the child, there is no information about typical symptoms for AS, these should be added.
  7. Line 184- should’t it be “by the mother”, not ”from the mother”?

Author Response

Dear Reviewers,

I would like to thank you for your valued comments and suggestions to the article. As you requested, we made all the necessary changes in our manuscript to address your concerns and we detailed below how the points raised have been accommodated. The main changes are written in red in the text of the manuscript. From the changes made in the revised manuscript and responses provided below, I hope you are convinced that we have adequately addressed the reviewer’s concerns and made the paper better. If there are any further questions, please feel free to let me know.

Manuscript concerned rare genetic syndromes and extremely rare possibility of giving birth by persons with Prader-Willi syndrome (PWS). This is interesting and good presented case report, which focused our attention on new aspect of care and sexual education patients with this syndrome. But I think, a few points should be more developed or added.

1. I think that genetics of PWS, and problem of maternal imprinting should be more explain in the introduction. PWS is human genetic imprinting disorder that can be caused by paternal deletion, maternal uniparental disomy or imprinting centre defect, not only this what was described.

Thanks for the suggestion. We better clarified the genetics of PWS. Please see lines 36-38

2. The way, how Angelman Syndrome (AS) develops should be showier expound.

Thanks for the suggestion. We deepened the causes and the characteristics of AS in the introduction, please see lines 31-35 and 48-52

3. There are too small about characteristic features of AS children.

Thanks for the suggestion. We reported additional data in our possession about the features of AS children. Please see lines 125-131.

4. Minor problems Line 31-32- I suggest replace the sentence from: “It has become clear that loss of function of the UGE3A gene, expressed on the maternal chromoseome, only causes AS” to “…………, expressed only on the maternal chromosome, only causes AS. What definitely change the meaning.

Thanks for the suggestion. We corrected as indicated.

5. I do not know, if it is necessary to write about PWS-like syndrome.

Thanks for the suggestion. We corrected as indicated

6. Lines 102-103- this sentence should be changed- trunk control is not the cause of GERD, what is suggested by this sentence.

Thanks for the suggestion. We rewrote the sentence,

7. Despite quite long follow up of the child, there is no information about typical symptoms for AS, these should be added.

Thanks for the suggestion. We reported additional data in our possession about the features of AS children. Please see lines 125-131

8. Line 184- should’t it be “by the mother”, not ”from the mother”?

Thanks for the suggestion. We corrected as indicated.

Round 2

Reviewer 1 Report

Lines 138 and 139:However, they have a premature pubarche characterized by earlier appearance of body hair, some breast development in girls and a degree of penile development in boys.

I would write: However, they have a premature pubarche characterized by earlier appearance of pubic hair. In girls there is mostly some breast development and in boys a degree of penile growth. (While like you put it, it reads like the breast development is part of the premature pubarche.)

In thet next sentences: 140 and 142: skip "phimosis" and skip " Labia minora" as these are not correct.

Author Response

Dear Reviewer,

We appreciate the time and effort that you dedicated to improve the quality of our manuscript and we are grateful for your valuable comments. We have incorporated all your suggestions.The changes are written in red in the text of the manuscript. If there are any further questions, please feel free to let me know.

Reviewer Comments:

Lines 138 and 139: However, they have a premature pubarche characterized by earlier appearance of body hair, some breast development in girls and a degree of penile development in boys.

I would write: However, they have a premature pubarche characterized by earlier appearance of pubic hair. In girls there is mostly some breast development and in boys a degree of penile growth. (While like you put it, it reads like the breast development is part of the premature pubarche.)

Thanks for the suggestion. I corrected as you suggested.

In thet next sentences: 140 and 142: skip "phimosis" and skip " Labia minora" as these are not correct.

Thanks for the suggestion. I skipped the terms as you suggested.